# Synthesis, Molecular Simulation, DFT, and Kinetic Study of Imidazotriazole-Based Thiazolidinone as Dual Inhibitor of Acetylcholinesterase and Butyrylcholinesterase Enzymes

**DOI:** 10.3390/ph18030415

**Published:** 2025-03-15

**Authors:** Manal M. Khowdiary, Shoaib Khan, Tayyiaba Iqbal, Wajid Rehman, Muhammad Bilal Khan, Mujaddad Ur Rehman, Zanib Fiaz

**Affiliations:** 1Department of Chemistry, Faculty of Applied Science, University College-Al Leith, University of Umm Al-Qura, Makkah 21955, Saudi Arabia; 2Department of Chemistry, Abbottabad University of Science and Technology, Abbottabad 22500, Pakistan; 3Department of Chemistry, Hazara Univeristy, Mansehra 21120, Pakistan; 4Department of Microbiology, Abbottabad University of Science and Technology, Abbottabad 22500, Pakistan; 5Department of Chemistry, Balochistan University of Information and Technology, Engineering & Management Sciences, Quetta 87300, Pakistan

**Keywords:** Alzheimer’s disease, triazole-thiazolidinone, molecular docking, pharmacophore modeling, MD simulations

## Abstract

**Background:** Alzheimer’s disease is a complex and multifactorial brain disorder characterized by gradual memory impairment, cognitive disturbance, and severe dementia, and, ultimately, its progression leads to patient death. This research work presents the design, synthesis, and characterization of novel imidazotriazole-based thiazolidinone derivatives (**1**–**14**), displaying promising anti-Alzheimer’s activity. **Methods:** These derivatives were synthesized by using 1*H*-imidazole-2-thiol as a starting reagent. Structural characterization was accomplished by ^13^C-NMR and ^1^H-NMR, while the molecular weight was confirmed by HREI-MS. These compounds were investigated for their anti-Alzheimer’s potential under an in vitro analysis. **Results:** These compounds showed a significant to moderate biological potential against AChE and BChE in comparison to donepezil (IC_50_ = 8.50 µM and 8.90 µM against AChE and BuChE), used as a reference drug. Among these compounds, analog **10** with IC_50_ values of 6.70 µM and 7.10 µM against AChE and BuChE emerged as the lead compound of the series with promising biological efficacy against targeted enzymes. Molecular docking revealed the interactive nature of active ligands against target enzymes. These compounds were also assessed under dynamic conditions to examine the structural deviation and conformational changes in a protein complex structure. DFT calculations provided the relative stability and reactivity of the lead compounds. An ADMET analysis showed that these compounds have no toxicological profile. **Conclusions:** This research study paves the way for the further development and optimization of novel and selective imidazotriazole-based thiazolidinone inhibitors as potent anti-Alzheimer’s agents.

## 1. Introduction

Alzheimer’s disease is a complex and multifactorial brain disorder characterized by gradual memory impairment, cognitive disturbance, and severe dementia, and, ultimately, its progression leads to patient death [1,2,3]. It is one of the leading causes of death in the old-age population due to the continuous and irreversible brain degeneration [4]. People with AD initially observe difficulties in thinking, apathy, delusions, a loss in decision-making ability, and gradual memory loss, and, with time, severe behavioral changes, extreme anxiety, depression, and a complete loss of cognitive capabilities [5,6]. Brain degeneration occurs due to a decline in the acetylcholine level, β-amyloid plaque formation, and neurofibril aggregation through the activity of acetlycholinestaerse (AChE) and butyrylcholinesterase (BChE) enzymes [7,8,9]. These enzymes are responsible for the hydrolytic cleavage of Acetylcholine, a neurotransmitter released by the cholinergic neuron, which helps in balancing cognitive functions, reducing its level and degeneration of cholinergic neurons by acetate produced in the hydrolysis reaction [10]. Different drug moieties were reported for the inhibition of these enzymes and to control brain degeneration. These mostly include tacrine, rivastigmine, and galantamine [11]. These drug candidates do not have a very effective biological profile due to their attached substituents, and they also have several side effects on the human body [12]. All these conditions emphasize the need for competitive inhibitors with a strong biological potential, highly interactive properties, stability, and very minimal or no side effects. In this regard, the imidazole and triazole moieties were fused and further merged with the thiazolidinone ring due to their biological significance to report novel hybrid imidazotriazole-based thiazolidinone derivatives (**1**–**14**). The imidazole ring is one of the important key structural elements used in different moieties as a precursor, which were used as anti-microbial [13], anti-Alzheimer’s [14], anti-inflammatory [15], and anti-cancer [16] agents. Similarly, the triazole moiety also has an excellent profile as anti-cancer [17], anti-microbial [18], anti-diabetic [19], and anti-inflammatory [20] agents. They were also used as a precursor molecule for the synthesis of several drug candidates. The thiazolidone ring is one of the significant heterocyclic moieties used in the core skeleton of many anti-inflammatory [21], anti-oxidant [22], anti-depression [23], anti-Alzheimer’s [24], and anti-diabetic [25] agents. Thus, to produce a synergetic effect of all moieties in a single hybrid and collectively enhance the potency of these components in a single molecule, novel imidazotriazole-based thiazolidinone derivatives were synthesized under optimized reaction conditions. Further, they were assessed in vitro against AChE and BChE enzymes, and, to correlate these findings with computational means, different in silico studies, including molecular docking, Pharmacophore modeling, MD simulations, DFT, ADMET, and enzyme kinetics, were performed.

Similarly, a rationale study is also illustrated in Figure 1 to compare the structure and the respective biological profile of newly synthesized compounds with previously reported analogs containing the triazole [26] and thiazolidinone [27] moiety in their core skeleton. This rationale study shows that the previously reported compounds containing the triazole and thiazolidinone moiety are less effective in inhibiting the enzymes. This reveals the high demand of anti-Alzheimer’s agents with a high efficacy and minimal side effects. For this purpose, we synthesized hybrid compounds of triazole and thiazolidinone, which have strong potential as anti-Alzheimer’s agents. These compounds were analyzed for their inhibition potential via an in vivo analysis. The outcomes of the in vivo study were also studied and validated under an in silico molecular docking study. All the in vivo and in vitro results of the current research study revealed that the lead compounds of the series have the potential to provide the highly necessary treatment for Alzheimer’s disease.

## 2. Results and Discussion

### 2.1. Chemistry

These imidazotriazole-based thiazolidinone derivatives were synthesized using an effective synthetic approach consisting of a four-step reaction. In this reaction, 1*H*-imidazole-2-thiol was used as a precursor molecule, which was allowed to react with hydrated hydrazine in an ethanol solvent under continuous heating and stirring conditions for about 5 h to obtain 2-hydrazinyl-1H-imidazole as an intermediate moiety. In the next step, this intermediate was cyclized by reacting with 4-amino-3-methoxybenzoic acid under the catalytic effect of acetic acid in ethanol to give the 4-(7H-imidazo[2,1-c][1,2,4]triazol-3-yl)-2-methoxyaniline moiety as the second intermediate. In third step, again in ethanol and acetic acid, this moiety was treated with a varied substitution containing benzaldehyde to afford imidazotriazole-based Schiff base derivatives. In the last step, a cyclization reaction occurred between the intermediate (III) and thioglycolic acid to obtain imidazotriazole-based thiazolidinone derivatives (as shown in Figure 1) under the same reaction conditions (ethanol, acetic acid and reflux for about 6 h). These derivatives have an appropriate yield and were further purified by washing with n-hexane. At the end, solvents were evaporated, and fine products were collected to further elucidate their structures via ^1^H-NMR and ^13^C-NMR. The molecular weight of these molecules was calculated to further confirm their synthesis by HREI-MS. In each step, product confirmation was achieved through single and clear spots on TLC plates. The melting point, percentage yield, retardation factor (R_f_) and color of novel synthesized imidazotriazole-based thiazolidinone derivatives (**1**–**14**) are given in Table 1.

### 2.2. Spectral Analysis

#### 2.2.1. Description of ^1^H-NMR

To elucidate the structure of synthesized compounds, ^1^H-NMR and ^13^C-NMR were conducted. The Advance spectrophotometer was employed with a radio frequency range of 600 and 150 MHz. DMSO-d_6_ was used as a reference solvent to dissolve these products and effectively conduct NMR. The ^1^H-NMR results for analog 10 have different peaks and multiplicity. In this regard, the proton of –NH in the imidazole ring is highly de-shielded and appears at a chemical shift value of 12.94 ppm, which gives a singlet. The proton of aromatic ring B attached to carbon 5′ has a chemical shift value of 7.75 ppm and, coupled with a neighboring proton, gives a doublet with a couple constant *J* = 7.25 Hz. On the third number, protons of C-2 and C-6 (Figure 2) appear at a chemical shift value of 7.52 ppm and give a doublet by coupling with the proton attached to nearby carbons. Both protons have peaks at the same position due to the same chemically and magnetically equivalent nature and have a coupling constant *J* = 7.35 Hz. These are de-shielded protons due to the strong electron-withdrawing power of the –CF_3_ moiety in the *ortho*-position. Next, a peak appeared for the proton of the aromatic ring attached to C-6′, giving a doublet with a chemical shift and coupling constant value of 7.42 ppm and 7.35 Hz, respectively. At 7.31 ppm, a peak of the proton attached to C-3′ appears, which gives a singlet. The next peak appears for protons of the substituted phenyl ring at 7.20 ppm, giving a doublet with a coupling constant *J* = 7.26 Hz. They are also de-shielded protons due to the –CF_3_ moiety. The next peak appears for protons of the imidazole ring at 7.14 ppm and 7.03 ppm, respectively. They also couple with each other and give a doublet with a coupling constant *J* = 7.23 and 7.20 Hz, respectively. The next peak appears for the proton between the nitrogen and sulfur atom, giving a singlet and chemical shift value of 6.42 ppm. The next peak appears for two protons of the thiazolidinone ring, attached to carbonyl carbon, and appears at 3.94 ppm. The last peak appears for aliphatic protons of the methyl group at 3.86 ppm.

#### 2.2.2. Description of ^13^C-NMR

In ^13^C-NMR, 21 different peaks were observed for each carbon in a molecule. Among them, the highly de-shielded carbon is a carbonyl carbon of the thiazolidinone ring (Figure 2), due to double-bonded highly electronegative oxygen atoms, and appears at 171.4 ppm. The next peak appears at 153.2 ppm and represents the C-2′ of the aromatic ring. At 152.4 ppm, the peak represents the common carbon of the imidazole and triazole ring. The next peak appears for carbon of the triazole ring at 152.1 ppm. It is also de-shielded carbon due to the electronegativity of both nitrogen atoms. At 142.3 ppm, the peak representing carbon 4 is also de-shielded due to the attached thiazolidinone ring. The next peak represents carbon of the imidazole ring attached to the -NH atom and appears at 135.4 ppm. The next peak represents C-1 129.3 ppm, while the next peak represents carbon 3 and 5 due to the attached –CF_3_ moiety and appears at 129.2 ppm, 129.1 ppm and 128.5 ppm, respectively. The next peak shows that carbon 4′ appears at 127.2 ppm. The next two peaks for C-2 and C-6 appear at 125.3 ppm and 125.2 ppm. At 124.2 ppm, the peak appears for carbon of the triflouromethyl group. The next peak represents carbon 6′, which appears at 123.1 ppm. The next peak shows the C-5′, imidazole carbon and C-3′ appears at 120.3 ppm, 114.4 ppm and 111.1 ppm, respectively. At 72.6 ppm, the peak represents carbon of the thiazolidinone ring between carbon and sulfur atoms. The last two peaks appear at 55.7 ppm and 33.3 ppm, which represent methoxy carbon and the remaining carbon of the thiazolidinone ring. In this way, the complete structure of compounds is elucidated by ^1^H-NMR and ^13^C-NMR, which gives information about the number of protons and carbon in molecules and their respective neighboring environment (attached substituents).

### 2.3. AChE and BChE Activity Analysis

Alzheimer’s disease is a chronic disorder characterized by an irreversible decline in the cholinergic system and cognitive functions due to AChE and BChE activity. So, the inhibition of these enzymes is one of the critical approaches to control neural degeneration and reduced neurotoxicity. These derivatives were found as inhibitors of these enzymes, with significant to moderate potency, having an inhibitory range of 6.70 to 21.40 µM for the AChE enzyme in comparison to donepezil (8.50 µM) and for BChE (7.10 to 22.10 ± 0.10 µM in contrast to donepezil drug with IC_50_ value of 8.90 ± µM). Fourteen different derivatives were synthesized and, among them, mostly flouro-substituted analogs hold maximum biological potential to their favorable interactions with targeted protein complexes (Table 2).

#### Structure–Activity Relationship (SAR)

Among these compounds, analog 10 has a strong biological profile with an inhibitory range of 6.70 and 7.10 µM, which overcomes the biological potency of donepezil (Figure 3 and Table 2). The imidazotriazole ring of analog 10 interacts with the enzyme active site through strong interactions, like hydrogen bonding and pi-pi stacking. The thiazolidinone ring also makes strong hydrogen bonds, resulting in strong binding between the ligand and enzyme active sites. The –CF_3_ moiety at the para position leads to hydrogen bonding using fluorine atoms and involves the ring in making cation interactions. The highly electronegative small-sized fluorine atoms cause less steric hindrance in a molecule. These characteristics give the molecule a stable spatial confirmation, and the ligand can easily fit into the binding sites.

Analog 3 also has a better biological profile (8.10 and 8.50 µM) than donepezil due to *ortho* and *para*-flouro substitutions. Both the imidazotriazole and thiazolidinone rings, along with the varied substituted phenyl ring and fluorine atoms, make strong forces of interactions to bind with the enzyme and reduce the substrate binding affinity with the targeted protein complex. The activated phenyl ring due to the electron-donating effects of fluorine atoms in the favorable position of attachment also makes several pi-pi-stacked and other polar interactions with electrophilic moieties in a targeted protein complex. Both flouro atoms lead to hydrogen bonding and cause confirmation rigidity in a molecule. It helps the ligand to better orient in a targeted protein complex without any major change in the optimal binding confirmations. It also increases the lipophilicity of a molecule and acts as a bioisostere, which leads to the compound’s ability to create new and stable interactions by maintaining its binding affinity. Moreover, analog 11 also exhibits significant biological potency (8.30 and 8.60 µM) and holds *ortho*-hydroxyl and *para*-flouro groups (Figure 3). –OH leads to hydrogen bonding interactions by both O and H atoms as well as increasing the nucleophilicity of the phenyl ring by donating a lone pair of electrons. This ring further leads to different interactions, mostly including pi-pi stacking with aromatic residues of targeted amino acids. Similarly, fluorine atoms increase the conformational rigidity of a molecule, producing hydrogen bonding and enhancing the lipophilic character. So, both –OH and –F moieties show synergetic effects and inhibit enzyme activity through strong interactions. Both substituents sterically hinder the enzyme active site and do not allow normal substrates to bind with enzyme active sites. When the hydroxyl position changes to the *meta*-position, its activity declines (10.40 and 10.90 µM), as observed in analog 12 (Table 2). In the *ortho*-position, the –OH moiety makes strong hydrogen bonds, but at the *meta*-position, its ability is reduced. Similarly, the nucleophilicity of the phenyl ring is also reduced to the minimum electron donation by the hydroxyl group at the *meta*-position. It is near the fluorine atom and might become involved in hydrogen bonding with it. So, this also sterically hinders the activity of the flouro group. The interactive properties of both groups were disturbed, which were observed in analog 11, due to a change in position. When the hydroxyl group is replaced with a nitro moiety, the potency is significantly reduced to 16.20 and 16.70 µM, as in analog 13. This might be due to the nitro moiety being involved in making interactions within a molecule and also giving a molecule structural confirmation, in such a way that it does not fit into the active sites of the targeted enzymes. Analog 9 shows somewhat moderate potential, as compared to donepezil, as it holds *meta*-substitutions of the flouro and hydroxyl moiety. In this case, both attached moieties are far from each other and have less steric hindrance, as compared to analog 12. So, they are capable of making a few binding interactions with the amino acids of the targeted enzymes.

Moreover, the inhibitory response of these derivatives against targeted enzymes was explored by graphical means. The data were illustrated in the form of a graph, which states the inhibition rate on the y-axis and the inhibitor concentration rate on the x-axis. The change in concentration leads to an enhanced inhibition rate, and the concentration at which the analog shows maximum inhibition is recognized as the IC_50_ value. The point of saturation also showed that, at this point, an increase in the inhibitor concentration has no effect on the inhibition rate or it remains the same. Figure 3 and Appendix A depict the drug–dose relationship for potent analogs against AChE and BChE complex.

### 2.4. Enzyme Kinetics

This is an in vitro approach, which measures the initial inhibition rate with the change in the inhibitor concentration. It explains the mechanism of action for how the inhibition rate varies with the passage of time by changing the inhibitor concentration, and the resulting curves or slopes show different competence levels of inhibitors, like a competitive inhibitor, uncompetitive inhibitor and non-competitive inhibitor. In this research study, analog 10 shows the maximum inhibition with strong binding affinity and was found as a competitive inhibitor. Its enzyme kinetic slopes show that they have a similar y-intercept and fight effectively to bind with the enzyme active site in comparison to the substrate binding affinity. The moderate biological profile of analog 9 leads to its uncompetitive inhibition, which is also illustrated through a kinetic study graph. It shows the number of parallel lines having different y-intercepts, and Km and Vmax gradually declined, which reduce the binding affinity of the inhibitor for the enzyme, as compared to a normal substrate. Similarly, poor inhibition or non-competitive inhibition were shown by analog 5 due to attached bulky substituents. The Km constant remains unchanged while having a different y-intercept. This inhibitor binds to the allosteric site of the enzyme and shows minimum inhibition.

Figure 4 shows the mode of action of analog 10 as a competitive inhibitor, while Figure 5 and Figure 6 show the inhibition action of analog 9 and 5 (as uncompetitive and non-competitive inhibitors).

### 2.5. Molecular Docking

Molecular docking is a kind of virtual screening, which is conducted by using different computational means (MOE, pymol and discovery studio visualizer) [28,29,30] to assess and examine the forces of interactions of active ligands with a targeted protein complex. This investigation involves the preparation, optimization and docking of ligands and targeted ligands to efficiently explore the binding affinity and their inhibitory efficacy. By using the Protein Data Bank as a source for retrieving the crystal structure of proteins, using their respective codes or PDB IDs (1POP code used for BChE and 1ACI for AChE enzyme), proteins were collected. Co-crystalline ligands were removed from the retrieved proteins, and active ligands were docked in the binding sites. The docking process was run through Gold docking tools, and poses with effective binding affinity were explored and critically visualized by pymol (Anaconda3 (64-bit)) software. The results show spellbinding outcomes with the number of interactions by active compounds with various amino acids of the targeted protein. These analogs have remarkable potential, and this was also confirmed by the molecular docking approach. Further, 2D visualization of the docking outcomes shows the interactions of analogs through different components of molecules having different binding distances, while 3D visualization shows the fitting of a ligand in a binding pocket or receptor site of the targeted enzyme complex, as shown in Figure 7 and Figure 8 as well as Table 3.

### 2.6. Pharmacophore Modeling

To further confirm the docking results and to better understand the interactive nature of the attached substituents responsible for the effective binding affinity of ligands in a targeted protein, Pharmacophore modeling was carried out [31]. This analysis critically evaluates and investigates the substituents, heteroatoms and each component of a molecule, which were involved in making strong and effective interactions, especially hydrogen bonding. The results show that these analogs have specific bioactivity profiles and effectively held in the binding pockets of the targeted protein complex, through different forces of attraction. The fluorine atoms of the –CF_3_ group, sulfur atoms and methyl group of methoxy were involved in strong hydrogen bond interactions, which were responsible for the bioactivity of analog 10, as shown in Figure 9.

### 2.7. MD Simulations

MD simulations are another kind of virtual analysis, which give information about the stability and structural deviation of the targeted protein complex, after docking the active ligand. A suitable simulated environment was provided for the protein structure and process run for 100 ns (simulations time). After running MD simulations, the root mean square deviation was observed to explore the stable interactions with the active ligand. The results show that the active ligands were effectively held in the binding pockets of the targeted protein complex (AChE), without any major root mean square fluctuations for around a 60 ns simulation time. This also shows that after docking analog 3 in the AChE enzyme, there are stable interactions and very few confirmation changes (structural deviation) in a dynamic environment. RMSF values were also assessed for active molecules in a targeted protein complex, and the results show that this active ligand has very few fluctuations and a root mean square deviation value less than 3 Å [32,33]. This shows that the ligand has stable and strong binding affinity in the binding site of the targeted protein complex. Figure 10 depicts that the simple protein and ligand–protein complex have the same behavior in a dynamic environment, as, up to 60–70 ns, there is no fluctuations or structural deviation in the protein complex structure. After 80 ns, it shows some fluctuations due to the interactions of ligands but, still, the ligand–protein complex has a strong stable character and does not show major structural conformational changes.

Similarly, the ion effects on potent analog **10** were also observed to determine their solvation properties, and the results sows they effectively interacted with different ions due to the attached substituents. The results are illustrated in Figure 11.

### 2.8. ADMET Profile

To assess the toxological profile of active compounds by ADMET predictions, the swissADME computational tool was employed. This analysis explains the mechanism of absorption, solubility, skin permeability, pharmacokinetics and medicinal properties, as well as their metabolic pathways, their excretion and the toxic character of active compounds. Under certain rules based on the molecular weight, the number of hydrogen bond donor and acceptor atoms, rotatory bonds, heavy atoms as well bioavailability, log Kp, Muegge, Egan, Veber and lead likeness violations were assessed, and the results showed that they did not exhibit any critical violations and have specific therapeutically safe characteristics. The ADMET profile proved that these compounds can be used as therapeutically safe drug candidates due to their drug-like attributes (as illustrated in Appendix A).

### 2.9. DFT

DFT is one of the computational approaches employed to study the behavior of molecules, their electronic structure and relative stability as well as reactive sites [34]. Molecule dynamics were explored by using a software package. For the optimization of derivatives, the ωB97XD functional was employed at the theoretical level, while vibrational modes and optimized structures were further confirmed by vibrational energy analysis at the potential energy surface (PES) [35,36]. Gauss View 5.0 and Chemcraft were used to visualize the DFT outcomes, revealing the distribution of electrostatic potential and formation of molecule orbitals [37]. MESP and FMO analyses are two different kinds of DFT approach, which efficiently explore the electronic arrangements and HOMO as well as LUMO orbitals and their relative energy gaps.

#### 2.9.1. MESP Analysis

MESP explores the distribution of the electrostatic potential of active compounds within a molecule to explore their reactive sites, which are susceptible to attacks of different moieties from the amino acids of targeted enzymes. A color-coded pattern was used to investigate the electrostatic potential of molecules and visualize the 3D arrangement of electronic sites, which were responsible for the strong interactions observed in molecular docking. This shows that all these have different electrostatic potential minima and maxima, depending upon the nature of substituents and heteroatoms in a molecule [38,39,40]. The MESP results showed that these analogs have a spatial arrangement of molecules and have uncleophillic as well as electrophilic characteristics due to the presence of a lone pair of electrons and the strong electron-withdrawing power of heteroatoms. All analogs have nucleophilic sites around the nitrogen atoms of the triazole ring and carbonyl oxygen of the thiazolidinone moiety due to high charge density. They have negative potential of −7.682 × 10^−2^ , indicated by the red color, and are attacked at electrophilic sites. Hydrogen atoms attached to nitrogen of the imidazole ring act as a highly electrophilic region in all analogs and have positive potential of 7.682 × 10^−2^ . Similarly, other hydrogen atoms of the substituted ring and imidazole moiety also hold positive potential, represented using light-blue color, as shown in Figure 12.

#### 2.9.2. FMO Analysis

This analysis explains the electronic density distribution between different components of molecules, which contributes to the overall potential of compounds as therapeutic agents. This analysis gives information about the molecular orbitals formed in a molecule and their respective energies [41]. Both HOMO and LUMO orbitals have specific energy, and, from their difference, energy gaps were calculated. This energy gap helps to determine the relative stability and reactivity of a molecule to bind with an enzyme complex and show maximum inhibition. In all analogs, LUMO orbitals have maximum lobe formation around the overall molecule due to the shifting of electronic density, while for HOMO orbitals, lobe formation occurs other than the substituted phenyl ring, as shown in Figure 13. The energy gap for analog 3 is −0.16513, 10 is −0.16141 and 11 is −0.16731.

## 3. Materials and Methods

### 3.1. Materials

Precursor molecule, 1*H*-imidazole-2-thiol, solvent ethanol, catalyst acetic acid and other reactants were bought from Sigma Aldrich and Merck (Hamburg, Germany) to carry out facile synthesis. Product confirmation was initially achieved through aluminum-containing silica plates and further by Advance bruker AM spectrophotometer, Spectroscopic techniques (^1^H-NMR and ^13^C-NMR) were conducted at frequencies of 600 and 150 MHz, respectively. This gives information about the basic skeleton of synthesized molecules by NMR spectra having various peaks and multiplicity. Each proton and carbon has different chemical shifts and coupling constant values measured in ppm and Hz. BuchiM-560 was used as boiling-point measuring apparatus. HREI-MS was employed (using Finnigan MAT-311A mass spectrometer, Finnigan MAT, San Jose, CA, USA) for measuring the molecular weight of the different molecular fragments of the synthesized molecules, to confirm their synthesis.

### 3.2. Methodology for the Synthesis of Imidazotriazole-Based Thiazolidinone Derivatives

These imidazotriazole-based thiazolidinone derivatives were synthesized by four-step reaction under mild reaction parameters. In this reaction, 2 mmol of 1*H*-imidazole-2-thiol was used as a precursor molecule, which was allowed to react with 2 mmol of hydrated hydrazine in ethanol (10 mL) solvent under continuous heating and stirring conditions for about 5 h to obtain 2-hydrazinyl-1H-imidazole as intermediate moiety.

In the next step, this intermediate 2-hydrazinyl-1H-imidazole (2 mmol) was cyclized by reacting with 2 mmol of 4-amino-3-methoxybenzoic acid under the catalytic effect of 3 drops of acetic acid. This reaction was carried out in 10 mL ethanol to obtain 4-(7H-imidazo[2,1-c][1,2,4]triazol-3-yl)-2-methoxyaniline moiety as second intermediate.

In the third step, 4-(7H-imidazo[2,1-c][1,2,4]triazol-3-yl)-2-methoxyaniline was treated with varied substituted benzaldehyde (each 2 mmol) in 10 mL ethanol and 3 drops of acetic acid to afford of imidazotriazole-based Schiff base derivatives.

In the last step, cyclization reaction was carried out by mixing imidazotriazole-based Schiff base derivatives (2 mmol) and thioglycolic acid to obtain novel series of imidazotriazole-based thiazolidinone derivatives under same reaction conditions (10 mL ethanol solvent, 3 drops acetic acid as catalyst and reflux for about 6 h). These derivatives have appropriate yield and were further purified by washing with n-hexane. At the end, solvents were evaporated, and fine products were collected to further elucidate their structures via ^1^H-NMR and ^13^C-NMR. Molecular weight of these molecules was calculated to further confirm their synthesis by HREI-MS. In each step, product confirmation was achieved through single and clear spots on TLC plates.

### 3.3. AChE and BuChE Assay Protocols

With some modifications, inhibitory potential against AChE and BuChE was investigated according to the reported protocol [42]. The total volume of the reaction mixture was kept at 100 µL, containing 60 µL of Na_2_HPO_4_ buffer, with a concentration of 50 mM and a pH of 7.7. Added test compound (well-**1**) with volume of 10 µL and a concentration of 0.5 mM followed by the addition of an enzyme of 10 µL (0.005 unit well-**1**). Pre-incubation of substances at 37 °C for 10 min was achieved. Reaction was started by adding 10 µL of 0.5 mM well-1 substrate (acetylthiocholine iodide**/** butyrylthiochloine chloride) followed by addition of 10 µL DTNB (0.5 mM well-**1**). Absorbance at 405 nm was measured after 15 min of incubation at 37 °C using the 96-well plate reader Synergy HT, BioTek, Shoreline, WA, USA. All experiments were performed with their respective controls in triplicate. Standard drug used was donepezil. The % inhibition was computed using the following equation:Inhibition (%) = Control − Test/control × 100

Control EZ-Fit Enzyme kinetics software (version: 13.4) (Perrella Scientific Inc., Amherst, MA, USA) was used for the calculation of IC_50_ values.

### 3.4. Molecular Docking Protocol

The crystal structure was retrieved from the Protein Data Bank (PDB) and, further, the structure was optimized by removing the water molecules, heteroatoms, and cofactors. Hydrogen bonds, missing atoms, and charges were computed. The synthesized pyrazolone-derived thiazolidinone-based chalcone scaffolds used in these docking studies were prepared and optimized using built and Ligand Preparation module implemented in Discovery Studio 2018 (Dassault Systemes BIOVIA, San Diego, CA, USA). For the purpose of docking, Gold docking tool was used; Ligand Preparation includes generating various tautomer’s, assigning bond orders and stereochemistry. Additionally, receptor grid was generated around the AChE and BuChE active sites by choosing centroid of complexed ligand (Montbretin A). The active site was defined with a radius of 12 Å around the Montbretin A binding site. Docking calculations were accomplished using Chem PLP scoring function [43].

### 3.5. DFT Assay Protocol

The geometric parameters and energies were computed using density functional theory at the ωB97XD level of theory, using the GAUSSIAN 98W package of the programs [44], on geometries that were optimized at CEP-631G basis set. The high basis set was chosen to detect the energies at a highly accurate level. The atomic charges were computed using the natural atomic orbital populations. The ωB97XD is the key word for the hybrid functional [45], which is a linear combination of the gradient functionals proposed by Becke [46] and Lee, Yang and Parr [47], together with the Hartree–Fock local exchange function [48]. UV spectra were recorded in Rigol, Ultra-3000 series in Enzymology and Fungal Biotechnology Lab, Faculty of Science, Zagazig University.

## 4. Conclusions

In the current study, the facile synthesis of imidazotriazole-based thiazolidinonde derivatives (**1**–**14**) was carried out. All the synthesized compounds were collected and characterized through spectroscopic analysis (^1^H-NMR, ^13^C-NMR spectroscopy and HREI-mass spectrometry). These compounds were assessed for their anti-Alzheimer’s potential by inhibiting AChE and BChE enzymes. Their biological inhibition potential was studied in comparison to the standard drug donepezil. Among the members of the series, analog 10 with the –CF_3_ moiety in the para-position was found to effectively bind with the enzyme active site and reduce the binding affinity of the normal substrate. Analog 10 was marked as an excellent inhibitor in the series, with substantial potential to replace donepezil for Alzheimer’s treatment. The inhibition mode of analog 10 was also confirmed by the enzyme kinetics, providing information about the change in the inhibition rate by varying the concentrations. In silico molecular docking was also employed to explore the binding modalities formed between active analogs and target enzymes. The outcomes of this study were also strengthened by employing Pharamcophore modeling. Protein–ligand interactions and protein and structural deviation in a dynamic environment were explored using MD simulation studies. Density functional theory analysis also provided insights into the distribution of electrostatic potential and relative stability as well as the reactive sites of active compounds. The toxological characteristics of the lead compounds in the series were predicted by means of ADMET analysis. All the in vitro and in silico studies of the current research work show that the excellent inhibitors of the novel series can substantially replace donepezil for Alzheimer’s treatment in the future.

## Data Availability

The original contributions presented in this study are included in the article/Appendix A.

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
