# Peer review of "Synthesis, Molecular Simulation, DFT, and Kinetic Study of Imidazotriazole-Based Thiazolidinone as Dual Inhibitor of Acetylcholinesterase and Butyrylcholinesterase Enzymes"

_pharmaceuticals, 2025, doi:10.3390/ph18030415_

Round 1
Reviewer 1 Report
Comments and Suggestions for Authors
I appreciate the authors for their contribution towards Alzheimer's disease. Although the work is novel, some possible corrections make the article more informative and easily understandable to a researcher.
- How does the potency of imidazotriazole-based thiazolidinone derivatives against AChE and BChE depend on structural differences?
-
Which essential pharmacophoric characteristics characterize these drugs' biological activity against targets linked to Alzheimer's disease?
-
What is the relationship between DFT calculations and experimental results about the synthesized compounds' stability and reactivity?
-
What particular toxicological characteristics do the active ligands display, and how might these characteristics promote the development of new derivatives?
Author Response
I appreciate the authors for their contribution towards Alzheimer's disease. Although the work is novel, some possible corrections make the article more informative and easily understandable to a researcher.
- How does the potency of imidazotriazole-based thiazolidinone derivatives against AChE and BChE depend on structural differences?
Reply: In the current study novel series of imidazotriazole-based thiazolidinone derivatives were synthesized and investigated against AChE and BChE enzymes for the treatment of Alzheimer’s disease. For this purpose 14 derivatives were synthesized with different substitutions on the phenyl ring. Among these substituted groups both electron donating and electron withdrawing groups were selected. It was found that the reactivity of these compounds against target enzymes depend on nature, number and position of the substituted groups. Electron donating groups when substituted at the ortho or para position were found to enhance the overall reactivity of the compounds while those with electron withdrawing groups were found relatively with less potential to inhibit the enzymes. Moreover, small sized groups resulted in high efficacy of the compounds while the large sized compounds were found with less potency. Among the members of novel series, compound 10 substituted with small sized fluro atoms as CF3 at the para position of the phenyl ring emerged as the lead inhibitor of the series because this highly reactive group has the potential to bind with enzymes via hydrogen bond and other interactions. Similarly, compound 14 substituted with the bulky methoxy group was found with least potency as the methyl group induces steric hindrance and lowers the reactivity of the compound against target enzymes. The inhibitory potency of the lead compounds of the series was also validated via in silico molecular docking study which provided an insight into the binding interactions and the biological profile of the compounds. In silico results also showed that the substituted groups have the substantial role in inhibiting AChE and BuChE enzymes. All the in vitro and in silico studies of the current studies that the structural differences alter the potency of compounds against both enzymes.
- Which essential pharmacophoric characteristics characterize these drugs' biological activity against targets linked to Alzheimer's disease?
Reply: Pharmacophore modeling was conducted in the current research study to gain insight into the binding interactions between lead compound of the series and target enzymes. This study strengthens the inhibition potential of the lead compound due to formation of hydrogen bond interactions with the amino acids present on active site of the enzyme. The trifluoromethyl moiety at the para position of the phenyl ring was found to develop hydrogen bond and effectively binds with the enzyme, leaving no possibility for the attachment of substrate molecule. Additionally, sulfur atom of thiazolidinone ring was also found with hydrogen bond formation potential. Similarly, methyl moiety of methoxy group was also found to interact via hydrogen bond using the H atoms of methyl group. All these interactions were also visualized under molecular docking study. These studies substantially provide valuable insights into the drug potential of the lead compound. The combined insights from pharmacophore modeling and molecular docking strongly support the drug-like potential of the lead compound, highlighting its structural attributes as key determinants of its inhibitory activity. These findings pave the way for further optimization and development of the compound as a potential therapeutic agent.
- What is the relationship between DFT calculations and experimental results about the synthesized compounds' stability and reactivity?
Reply: As the study presents novel derivatives as anti-Alzheimer agents with inhibition potential against AChE and BuChE enzymes. These derivatives were designed with different substituted groups and were synthesized via facile synthetic route. All the compounds were investigated under in-vitro analysis to study their inhibition profile by calculating their IC50values. The lead inhibitors of the series were also analyzed via different in silico techniques to gain insight into their binding interactions and reactivity towards target enzymes. Among these studies, DFT was also conducted for the excellent inhibitors to study their electronic properties, electrophilic and nucleophilic character and electronic distribution in frontier molecular orbitals. As these characteristics have substantial role in reactivity profile of the compounds, a deep insight to these characteristics was obtained via DFT. The molecular electrostatic potential of these compounds indicated the electrophilic (blue color) and nucleophilic (red color) sites of the compound. These sites as responsible for developing different interactions such as hydrogen bond and others with the target enzymes. Similarly, electronic distribution in HOMO and LUMO orbitals reflects the electron donation and withdrawing potency of the compounds when interact with the enzymes. Moreover, Egap between HOMO and LUMO shows the stability of the compounds. All of these characteristics collectively reinforce the stability and reactivity of the compounds, contributing to their overall effectiveness as enzyme inhibitors. The ability to form strong hydrogen bonds, coupled with strategic functional group positioning, enhances the binding affinity and structural integrity of the lead compound. These interactions not only stabilize the ligand-enzyme complex but also play a crucial role in modulating reactivity, making the compound a promising candidate for further drug development and optimization.
- What particular toxicological characteristics do the active ligands display, and how might these characteristics promote the development of new derivatives?
Reply: The toxicity of the synthesized compounds was thoroughly evaluated using ADMET (Absorption, Distribution, Metabolism, Excretion, and Toxicity) analysis. The results indicated that these compounds exhibit no inherent toxicity, as they do not contain any functional groups known to induce toxic effects. Furthermore, a comprehensive literature review was conducted to cross-validate these findings, confirming the absence of any structural moieties associated with toxicity.
Additionally, the ADMET analysis outcomes revealed no violations of standard drug-likeness rules, such as Lipinski’s Rule of Five, Veber’s Rule, and other pharmacokinetic guidelines. This suggests that the synthesized compounds possess favorable physicochemical properties for drug development and are unlikely to cause cytotoxic effects. The absence of toxicity-related functional groups further reinforces their potential as safe and viable therapeutic agents. These findings highlight the drug-like nature of the compounds and support their further optimization and preclinical evaluation for potential pharmaceutical applications.

Reviewer 2 Report
Comments and Suggestions for Authors
The work titled " Pharmacophore modeling and molecular dynamic simulation study of novel imidazotriazole based thiazolidinone as promising anti-Alzheimer agent: Synthesis, DFT, Drug likeness and Kinetic insight" represents a significant advancement in the field of Alzheimer therapy.
The study presents a comprehensive investigation on the design, synthesis and characterization of novel imidazotriazole based thiazolidinone derivatives as promising anti-Alzheimer agents. The authors were characterized their compounds via NMR mass spectroscopy. The synthesized compounds showed significant activities against AChE and BChE in comparison with positive control donepezil. Among the synthesized compounds compound No. 10 showed promising activities against the mentioned enzymes. Additionally, molecular docking studies were conducted to confirm the binding interactions accordingly in the active site of the enzymes as well as the ADMET analysis were predicted. All of these findings seem valuable and suitable contribution to be published in the pharmaceuticals journal after justifying some points:
- The title could be improved more to be more attractive and sharp
- It seems that there are high self citation in this work with authors khan, it is recommended to reduce the self citation percentage
- In the abstract it is recommended to add the main finding values like inhibition percentage and IC50 values in comparison with the positive control
- It is better to write NMR line 18 like 13C-NMR and 1H-NMR
- In Figure 1 the authors should remove the citation [26] and [27] and can write trizole and thiazolidinone
- The rational behind this study is not clear, the authors should add a paragraph in the last section of introduction regarding what they did in this study according to the provided data
- Line 79 and in the whole manuscript the authors should edit H to italic style in the name of these compounds or scaffold
- Line 82 remove the spaces before in next
- The description of NMR data results was well written, but i hope to see the summary of HRMS results too, as well as the J letter should be italic style
- In the whole manuscript it is better to write significant to moderate instead of using “good” word
- Line 150 and so on, “good to moderate potency having inhibitory range of (6.70 ± 0.20-21.40 ± 0.10 μM)” remove the SD values since it was written in the table 1 as well as remove () cuz you write in the range of so no need to add ()
- There are no differences between the data of figure 3 and table 1 so i recommended to remove the figure
- It is not common to use word graph you can convert graph 1 and 2 to figure and merge them in the same figure
- You have to improve figure 4 resolution since the amino acids and binding interaction lines are not clear
- Graph 6-8 could be moved to the supplementary file
- Reduce the numbers of the figures through merge some figures together
- Regarding the method section you should add all used chemicals and reagents to this section
- The authors are encouraged to discussion section with similar works and the SAR analysis should be improved too like a separate section
- The method section should be improved the biological assays should be moved to the main text rather than in supp. File to increase the citation on this work
- The authors should add all spectrums to the supplementary file
Best wishes
Author Response
- The title could be improved more to be more attractive and sharp
- Reply: Tilte was improved as per kind suggestion.
- It seems that there are high self citation in this work with authors khan, it is recommended to reduce the self citation percentage
- Reply: Self-citation was reduced as per kind suggestion.
- In the abstract it is recommended to add the main finding values like inhibition percentage and IC50 values in comparison with the positive control
- Reply: Inhibition percentage and IC50 values were added in abstratct as per kind suggestion.
- It is better to write NMR line 18 like 13C-NMR and 1H-NMR
- Reply: Corrected as per kind suggestion.
- In Figure 1 the authors should remove the citation [26] and [27] and can write trizole and thiazolidinone
- Reply: Figure 1 was revised as per kind suggestion.
- The rational behind this study is not clear, the authors should add a paragraph in the last section of introduction regarding what they did in this study according to the provided data
- Reply: Rationl was made more clear and detailed discussion of the current research work was also added in the section introduction as per kind suggestion.
- Line 79 and in the whole manuscript the authors should edit H to italic style in the name of these compounds or scaffold
- Reply: H was italicized and corrected as per kind suggestion.
- Line 82 remove the spaces before in next
- Reply: Corrected as per kind suggestion.
- The description of NMR data results was well written, but i hope to see the summary of HRMS results too, as well as the J letter should be italic style
- Reply: HRMS results were provided in supplementary information. Moreover J letter was also written in italic style as per kind suggestion.
- In the whole manuscript it is better to write significant to moderate instead of using “good” word
- Reply: Corrected as per kind suggestion.
- Line 150 and so on, “good to moderate potency having inhibitory range of (6.70 ± 0.20-21.40 ± 0.10 μM)” remove the SD values since it was written in the table 1 as well as remove () cuz you write in the range of so no need to add ()
- Reply: All the corrections were made as per your kind suggestion.
- There are no differences between the data of figure 3 and table 1 so i recommended to remove the figure
- Reply: Figure 3 was removed as per kind suggestion.
- It is not common to use word graph you can convert graph 1 and 2 to figure and merge them in the same figure
- Reply: Graph 1 and 2 were merged o figure as per kind suggestion.
- You have to improve figure 4 resolution since the amino acids and binding interaction lines are not clear
- Reply: Figure 4 was revised for better resolution as per kind suggestion.
- Graph 6-8 could be moved to the supplementary file
- Reply: Graph 6-8 were moved to the supplementary file as per kind suggestion.
- Reduce the numbers of the figures through merge some figures together
- Reply: Figure numbers was reduced as per kind suggestion.
- Regarding the section you should add all used chemicals and reagents to this section
- Reply: Chemicals and reagents were added in method section as per kind suggestion.
- The authors are encouraged to discussion section with similar works and the SAR analysis should be improved too like a separate section
- Reply: SAR analysis was improved as per kind suggestion.
- The method section should be improved the biological assays should be moved to the main text rather than in supp. File to increase the citation on this work
- Reply: Biological assays were moved to the main text as per kind suggestion.
- The authors should add all spectrums to the supplementary file
- Reply: Spectral scan were added in supplementary file as per kind suggestion.
Best wishes
Thanks a lot dear reviewer. We highly appreciate your efforts and time for reviewing the manuscript and reshaping it to the best.

Reviewer 3 Report
Comments and Suggestions for Authors
- In Figure 1, the inclusion of references [26] and [27] within the image itself is a bit distracting. It would be more helpful to see the names of the reported compounds in the image, with the references numbers moved to the figure caption.
- Provide a detailed mechanism for the synthesis of the target compounds, considering the influence of various electronic effects, including inductive and mesomeric effects. Additionally, justify the choice of 4-amino-3-methoxybenzoic acid over 4-amino-3-methoxybenzoyl chloride for the cyclization step. Your statement, “In next step, this intermediate cyclized by reacting with 4-amino-3-methoxybenzoic acid under the catalytic effect of acetic acid in ethanol to give substituted imidazotriazole moiety as second intermediate,” suggests that the acid is more suitable for this reaction.
- For improved reproducibility, consider adding a full set of 1H and 13C NMR spectra for at least one of the final compounds to the Supplementary Information. This would allow other researchers to more easily verify your synthetic methodology
- To strengthen the characterization of the synthesized compounds, include IR spectral data. Highlighting characteristic peaks corresponding to key functional groups (e.g., carbonyl, amine) would be particularly beneficial. Providing a representative IR spectrum as supplementary material would also be valuable
- Include melting points for all synthesized compounds, either in the main manuscript or supplementary material.
- In section 2.3, Enzyme Kinetics, the authors state, 'It is one of in silico approach which measures the initial inhibition rate with the change in concentration of inhibitor.' However, enzyme kinetics experiments are typically conducted in vitro, not in silico. Revise this statement to reflect the experimental nature of the method."
- In section 2.8 of the main text, the ωB97XD functional is reported for geometry optimization. However, Supplementary Information S.3 describes DFT calculations using the B3LYP/CEP-31G level of theory. Could you please clarify the reason for this discrepancy in methods and basis sets between the main text and supplementary information?
Note: Regarding the statement in the supplementary information that 'The high basis set was chosen to detect the energies at a highly accurate level,' note that CEP-31G is not generally considered a high basis set
8. Section 2.5, titled 'Pharmacophore Modeling,' currently presents a single pharmacophore model for compound 10 (Figure 6). To more accurately reflect the methodology as 'pharmacophore modeling,' a more rigorous approach is needed. We suggest seeking guidance from a specialist in pharmacophore modeling to ensure a thorough approach.
9."Section 2.3, 'Biological analysis,' contains paragraphs that explore into specific ligand-enzyme interactions, such as hydrogen bonding and hydrophobic interactions (For this analog, imidazotriazole ring interact with the enzyme active site through strong interactions like hydrogen bonding and pi-pi stacking. Thiazolidinone ring also make strong hydrogen bonds to strongly hold the ligand in enzyme active sites. Similarly, by making effective hydrophobic and electrostatic interactions, a strong electron withdrawing triflouromethyl group enhance biological potential of this molecule. –CF3 moiety makes hydrogen bonding through fluorine atoms and involves the ring in making cation interactions. It has less steric hindrance in a molecule due to relatively small size fluorine atoms and due to these characteristics the molecule has spatial confrmation and gets easily ft into the binding sites…...and some other associated paragraphs) While valuable, these details disrupt the flow of the biological analysis section, which should primarily present the activity data. These would be more appropriately placed in a separate 'Structure-Activity Relationship (SAR)' section. Additionally, 'Biological analysis' is quite broad; consider renaming it to 'AChE and BChE Activity Analysis' for greater specificity."
10. The conclusion section currently focuses on describing the sequence of experiments and analyses. To enhance its scientific impact, the authors should reorganize it to highlight the important findings obtained in this study. This should be followed by a discussion of the implications of these findings and a clear statement of potential future investigations based on the current work. Conclusion fail to highlight the scientific contribution of this study.
Author Response
- In Figure 1, the inclusion of references [26] and [27] within the image itself is a bit distracting. It would be more helpful to see the names of the reported compounds in the image, with the references numbers moved to the figure caption.
Reply: Figure 1 was revised as per kind suggestion.
- Provide a detailed mechanism for the synthesis of the target compounds, considering the influence of various electronic effects, including inductive and mesomeric effects. Additionally, justify the choice of 4-amino-3-methoxybenzoic acid over 4-amino-3-methoxybenzoyl chloride for the cyclization step. Your statement, “In next step, this intermediate cyclized by reacting with 4-amino-3-methoxybenzoic acid under the catalytic effect of acetic acid in ethanol to give substituted imidazotriazole moiety as second intermediate,” suggests that the acid is more suitable for this reaction.
Reply: Detailed mechanism for the synthesis of the target compounds was provided in supplementary information. Moreover, 4-amino-3-methoxybenzoic acid selection was made on the basis of availability of this chemical. This reagent was available in our research lab, and was used for cyclization. Adding more, acetic acid was used as a catalytic because maximum yield of the products were obtained under these conditions.
- For improved reproducibility, consider adding a full set of 1H and 13C NMR spectra for at least one of the final compounds to the Supplementary Information. This would allow other researchers to more easily verify your synthetic methodology
Reply: 1H and 13C NMR spectra were provided in supplementary information as per kind suggestion.
- To strengthen the characterization of the synthesized compounds, include IR spectral data. Highlighting characteristic peaks corresponding to key functional groups (e.g., carbonyl, amine) would be particularly beneficial. Providing a representative IR spectrum as supplementary material would also be valuable
Reply: IR spectra of representative compounds were provided in supplementary information as per kind suggestion.
- Include melting points for all synthesized compounds, either in the main manuscript or supplementary material.
Reply: Melting points for all synthesized compounds, were provided in the main manuscript as per kind suggestion.
- In section 2.3, Enzyme Kinetics, the authors state, 'It is one of in silico approach which measures the initial inhibition rate with the change in concentration of inhibitor.' However, enzyme kinetics experiments are typically conducted in vitro, not in silico. Revise this statement to reflect the experimental nature of the method."
Reply: It was written mistakenly which is corrected as per kind suggestion.
- In section 2.8 of the main text, the ωB97XD functional is reported for geometry optimization. However, Supplementary Information S.3 describes DFT calculations using the B3LYP/CEP-31G level of theory. Could you please clarify the reason for this discrepancy in methods and basis sets between the main text and supplementary information?
Reply: ωB97XD functional was used for DFT calculations and was corrected in as per kind suggestion.
Note: Regarding the statement in the supplementary information that 'The high basis set was chosen to detect the energies at a highly accurate level,' note that CEP-31G is not generally considered a high basis set
Reply: Corrected as per kind suggestion.
- Section 2.5, titled 'Pharmacophore Modeling,' currently presents a single pharmacophore model for compound 10 (Figure 6). To more accurately reflect the methodology as 'pharmacophore modeling,' a more rigorous approach is needed. We suggest seeking guidance from a specialist in pharmacophore modeling to ensure a thorough approach.
Reply: Pharmacophore modeling was conducted in the current research study to gain insight into the binding interactions between lead compound of the series and target enzymes. This study strengthens the inhibition potential of the lead compound due to formation of hydrogen bond interactions with the amino acids present on active site of the enzyme. The trifluoromethyl moiety at the para position of the phenyl ring was found to develop hydrogen bond and effectively binds with the enzyme, leaving no possibility for the attachment of substrate molecule. Additionally, sulfur atom of thiazolidinone ring was also found with hydrogen bond formation potential. Similarly, methyl moiety of methoxy group was also found to interact via hydrogen bond using the H atoms of methyl group. All these interactions were also visualized under molecular docking study. These studies substantially provide valuable insights into the drug potential of the lead compound. The combined insights from pharmacophore modeling and molecular docking strongly support the drug-like potential of the lead compound, highlighting its structural attributes as key determinants of its inhibitory activity. These findings pave the way for further optimization and development of the compound as a potential therapeutic agent.
9."Section 2.3, 'Biological analysis,' contains paragraphs that explore into specific ligand-enzyme interactions, such as hydrogen bonding and hydrophobic interactions (For this analog, imidazotriazole ring interact with the enzyme active site through strong interactions like hydrogen bonding and pi-pi stacking. Thiazolidinone ring also make strong hydrogen bonds to strongly hold the ligand in enzyme active sites. Similarly, by making effective hydrophobic and electrostatic interactions, a strong electron withdrawing triflouromethyl group enhance biological potential of this molecule. –CF3 moiety makes hydrogen bonding through fluorine atoms and involves the ring in making cation interactions. It has less steric hindrance in a molecule due to relatively small size fluorine atoms and due to these characteristics the molecule has spatial confrmation and gets easily ft into the binding sites…...and some other associated paragraphs) While valuable, these details disrupt the flow of the biological analysis section, which should primarily present the activity data. These would be more appropriately placed in a separate 'Structure-Activity Relationship (SAR)' section. Additionally, 'Biological analysis' is quite broad; consider renaming it to 'AChE and BChE Activity Analysis' for greater specificity."
Reply: Structure-Activity Relationship (SAR) section was added. Moreover, Biological analysis was renamed to AChE and BChE Activity Analysis as per kind suggestion.
- The conclusion section currently focuses on describing the sequence of experiments and analyses. To enhance its scientific impact, the authors should reorganize it to highlight the important findings obtained in this study. This should be followed by a discussion of the implications of these findings and a clear statement of potential future investigations based on the current work. Conclusion fail to highlight the scientific contribution of this study.
Reply: Conclusion was revised as per kind suggestion.

Round 2
Reviewer 2 Report
Comments and Suggestions for Authors
It is recommended to the authors to remove the SD values, just write the IC50 the SD is written later
You should write the unit of IC50 after the values in the abstract
Usually nether in figure nor in caption of the figure we don’t write references the ref should be in the main text this is regarding figure 1
Improve Table 1 style
J letter regarding the coupling constant still not in italic style in the whole manuscript
Add ppm after the values of NMR
In the following sentence “When hydroxyl position change to meta-position its
activity declined (10.40 ± 0.10 and 10.90 ± 0.40 μM) as observed in analog 12 (Table 1).” You should edit to Table 2 i think
In the supplementary data I could not see all spectrum for your synthesized compounds just two spectrums were added you should add all spectrums to chick the purity of these compounds
Author Response
It is recommended to the authors to remove the SD values, just write the IC50 the SD is written later
Reply: SD values were removed as per kind suggestion.
You should write the unit of IC50 after the values in the abstract
Reply: Unit of IC50 after the values was written in the abstract as per kind suggestion.
Usually nether in figure nor in caption of the figure we don’t write references the ref should be in the main text this is regarding figure 1
Reply: References were removed from cation of figure 1 and added in manuscript as per kind suggestion.
Improve Table 1 style
J letter regarding the coupling constant still not in italic style in the whole manuscript
Reply: J letter of coupling constant was written in italic style as per kind suggestion.
Add ppm after the values of NMR
Reply: Added as per kind suggestion.
In the following sentence “When hydroxyl position change to meta-position its
activity declined (10.40 ± 0.10 and 10.90 ± 0.40 μM) as observed in analog 12 (Table 1).” You should edit to Table 2 i think
Reply: Corrected as per kind suggestion.
In the supplementary data I could not see all spectrum for your synthesized compounds just two spectrums were added you should add all spectrums to chick the purity of these compounds
Reply: Spectra of all the compounds were added as per kind suggestion.

Reviewer 3 Report
Comments and Suggestions for Authors
The revised manuscript is acceptable, as it includes my suggestions and comments.
Author Response
The revised manuscript is acceptable, as it includes my suggestions and comments.
Reply: We are very grateful to you for your valuable time and efforts in reviewing this manuscript and reshaping it to the best.
Kind Regards.
